# Cell Technologies in the Stress Urinary Incontinence Correction

**DOI:** 10.3390/biomedicines10020309

**Published:** 2022-01-28

**Authors:** Igor Maiborodin, Gennadiy Yarin, Sergey Marchukov, Aleksandra Pichigina, Galina Lapii, Sergey Krasil’nikov, Svetlana Senchukova, Maxim Ryaguzov, Inna Vilgelmi, Maksim Bakarev, Vitalina Maiborodina

**Affiliations:** 1Institute of Molecular Pathology and Pathomorphology, Federal State Budget Scientific Institution “Federal Research Center of Fundamental and Translational Medicine” of the Ministry of Science and Higher Education of the Russian Federation, Timakova st., 2, 630117 Novosibirsk, Russia; msv-1981@mail.ru (S.M.); apichigina@yandex.ru (A.P.); galapii@frcftm.ru (G.L.); professorkrasilnikov@rambler.ru (S.K.); senchukova@ngs.ru (S.S.); mbakarev@gmail.com (M.B.); mai_@mail.ru (V.M.); 2The Center of New Medical Technologies, Institute of Chemical Biology and Fundamental Medicine, The Russian Academy of Sciences, Siberian Branch, akademika Lavrenteva str., 8, 630090 Novosibirsk, Russia; metrogyl@yandex.ru (G.Y.); willis@ngs.ru (M.R.); dr.vilgelmi.allsurg@yandex.ru (I.V.)

**Keywords:** cell therapy, multipotent stromal cells, urinary incontinence

## Abstract

The scientific literature of recent years contains a lot of data about using multipotent stromal cells (MSCs) for urinary incontinence correction. Despite this, the ideal treatment method for urinary incontinence has not yet been created. The cell therapy results in patients and experimental animals with incontinence have shown promising results, but the procedures require further optimization, and more research is needed to focus on the clinical phase. The MSC use appears to be a feasible, safe, and effective method of treatment for patients with urinary incontinence. However, the best mode for application of cell technology is still under study. Most clinical investigations have been performed on only a few patients and during rather short follow-up periods, which, together with an incomplete knowledge of the mechanisms of MSC action, does not make it possible for their widespread implementation. The technical details regarding the MSC application remain to be identified in more rigorous preclinical and clinical trials.

## 1. Introduction

Pelvic organ prolapse and stress urinary incontinence are reported in 40–50% of postmenopausal women, affecting 200 million people worldwide. Stress urinary incontinence is more common in women after vaginal delivery than in patients who have undergone a caesarean section, this suggesting a lack of complete tissue repair after vaginal delivery. Standard therapies often provide symptomatic relief, but do not target against the underlying etiology, and exhibit tremendous patient-to-patient variability of results, limited success, and procedure-related complications. More clinical trials of new treatment methods involving the required number of patients and with evaluation the long-term results of therapeutic methods for the incontinence correction should be encouraged [1,2,3,4,5,6,7].

When studying the pathogenesis of stress urinary incontinence in women, it is necessary to pay attention to age-related changes of the female urethra. The minimum and maximum indicators of the length, width, area or volume of organs and structures in the lower urinary tract can normally vary up to 2–3 times. With age, in healthy women the absolute and relative length of the urethra, the urethrovesical angle, and the inclination of the urethra do not change. Both smooth and striated muscle tissues, which are part of various departments of the female urethra, undergo atrophy during the aging process. Smooth muscle tissue is less variable with age, but striated muscle symplasts are sometimes completely absent in urethral biopsies from elderly patients. With age, the vascularization and density of the innervation decrease in the urethral structures, but the content of connective tissue in the external urethral sphincter increases. Urinary tract mobility at young women is more pronounced than at older women [8].

Over the past two decades, tissue engineering and regenerative medicine have made significant advances. Although the term “regenerative medicine” covers most areas of medicine, in fact urology is one of the most progressive. A lot of urological innovations and inventions have been studied over the past decades. Given the quality-of-life problems associated with urinary incontinence, there is a strong incentive for the development and implementation of new technologies. In addition, there is potential for further significant progress in regenerative medicine approaches using biomaterials, multipotent stromal cells (MSCs), or combinations thereof. All this is based on the need to replenish anatomical or physiological tissue deficiencies, reduce morbidity, and improve the long-term effectiveness of treatment. The ideal material for these purposes must meet the following criteria: provide mechanical and structural support; be biocompatible and maintain consistency with surrounding tissues; be “suitable for purpose” to meet specific application needs ranging from static support to transmission of biologically active cell signals [9].

Along with MSC therapy, the rapidly expanding field of tissue engineering has a promising future in urology clinics. The renal tissue functional biounit has been developed using cellular technologies. Urinary excretion has been successfully demonstrated by embryonic kidneys generated from MSCs. It has been shown that artificial embryonic cells derived from pluripotent mouse stromal cells give rise to living offspring. Cell therapy represents an attractive alternative for the treatment of stress urinary incontinence: Myoblast and fibroblast therapy has been used safely and effectively. In addition, stress urinary incontinence has been successfully treated clinically with MSCs derived from muscle tissue. Skeletal muscle-derived MSCs differentiated into smooth muscle cells when implanted into the corpora cavernosa in experimental models. Various types of MSCs have been investigated for use in the repair of the external sphincter and striated muscle tissue of the urethra. The use of MSCs appears to be a feasible and safe method with promising results for treatment of patients with incontinence [4,6,10,11,12].

MSCs have the ability for self-renewal and differentiation into a number of cell types and promote the release of chemokines and the migration of cells necessary for tissue regeneration. Mesenchymal MSCs are progenitor cells with an increased ability to proliferate and differentiate and these cells are less tumorigenic than MSCs from adult tissues [13].

In connection with the above, the aim of the investigation was to make a narrative overview of the literature regarding experimental and clinical data with using cell technologies, especially MSCs, to improve the treatment of stress urinary incontinence and to facilitate the search of guidance for further research.

## 2. In Vitro Data Testifying the Promise of Using MSCs for the Urinary Incontinence Correction

Cellular therapy shall have the potential to offer future solutions for both the initial placement of a slings and the procedure failure treatment. Preclinical studies show that MSCs can enhance the repair of damaged tissue, either through direct integration and replacement of damaged tissue (differentiation), or through secreted factors that influence the recipient’s response mechanisms (paracrine effect) [14]. Sprague–Dawley rat adipose tissue MSCs were adsorbed onto polyglycolic acid fibers, which formed a scaffold with a shape that mimics a sling complex. The results demonstrated that tissue scraping may contain MSC after 12 weeks in vitro culture under static stress. With increasing culture time, the engineered tissue showed significant improvement in biomechanical properties, including maximum load and Young’s modulus, as well as mature structures of tissue collagen. In addition, differentiated MSCs cultured under static stress retained myoblast phenotype on polyglycolic acid scaffolds [15].

The type I collagen content during stress urinary incontinence in women is significantly reduced in the periurethral tissues at the level of the vagina and in its fibroblasts. Exosomes increased the expression of the col1a1 gene in these cells, the expression levels of TIMP-1 (endogenous or tissue metalloproteinase inhibitor) and TIMP-3 were upregulated in them with significant downregulation of MMP-1 (matrix metalloproteinase) and the level of MMP-2 expression. That is, the use of MSC exosomes increases the type I collagen content by increasing its synthesis and decreasing degradation by fibroblasts [16].

Thus, experimental studies have demonstrated that MSCs increase the content of collagen in the periurethral tissues and persist for a long time in conditions of static stress on several materials used for the manufacture of slings. This may be of relevance for future treatment modalities of stress urinary incontinence including the use of cellular technologies.

## 3. Cellular Technologies in the Treatment of Patients with Urinary Incontinence

The scientific literature of recent years contains a large amount of data devoted to the study of mesh structures and the possibilities of their modification using MSCs for implantation into patients for correcting tissue defects and pelvic organ prolapse. Based on the literary analysis, Maiborodin et al. [17] studied the influence of cellular technologies on the results of implantation of mesh materials used in urology. The authors conclude that the ideal implant has not yet been created. Additional studies with a longer follow-up period are needed to determine the most successful and safe methods and materials for the restoration of pathologically altered or lost tissues and the transition to clinical trials. It is also yet to come to an unambiguous understanding of the best sources of MSCs, ways for stimulation of proliferation, preservation, and delivery of these cells into the necessary tissues of the body, to thoroughly study the causes of inefficiency and the risks of developing various complications, especially in the long term. 

Injection therapy with formulations including MSCs has been developed as a minimally invasive alternative to the surgical treatment of stress incontinence. MSC treatment is believed to promote functional regeneration of the urethral sphincter in patients with suspected internal sphincter system insufficiency. Evidently autologous fat and muscle tissues appear to be the most suitable source of MSCs for urological applications [18].

Current published literature presents safety and efficacy data regarding adult autologous muscle-derived cell injection for urinary sphincter regeneration in 80 patients at 12-month follow-up. In these studies, no long-term adverse events were reported and patients undergoing cellular injection at higher doses revealed at least 50% reduction in stress leaks and pad weight at 12-month follow-up. All dose groups demonstrated statistically significant improvement in patient-reported incontinence-specific quality-of-life scores at 12-month follow-up. Most likely, injection of muscle-derived cells across a range of dosages are safe. Efficacy data suggest a dose–response with more patients responsive to the higher doses of these cells [19].

Pathology (insufficiency) of the ligamentous apparatus of the urethra is most often the main cause in the development of the stress urinary incontinence in women. The transplantation of autologous MSCs derived from adipose tissue into the periurethral region is a new treatment method for stress urinary incontinence. Ten women with symptoms of stress incontinence were injected with MSCs via a transurethral and transvaginal approach under urethroscopy observation. Urinary incontinence decreased significantly during the first 2, 6, and 24 weeks after injection therapy. Autologous MSC periurethral injection represents a safe but short-term effective treatment for stress urinary incontinence. However, further studies with a longer follow-up period shall be needed to confirm long-term effectiveness [1]. According to the data of Zambon et al. [5], cure and relapse rates were 40% and 70%, respectively, 1 year after incontinence therapy with periurethral injection of autologous adipose or muscle MSCs.

The results of a small uncontrolled single center clinical study showed effectiveness of injection of MSCs into the region of urethral sphincters for the stress urinary incontinence correction. These results should be confirmed in larger cohort and controlled studies with longer follow-up that also evaluates applicability and safety.

## 4. MSC Incontinence Correction in Experimental Models

The regenerative potential of MSCs derived from the human dental pulp was evaluated in an animal model of stress urinary incontinence. To simulate stress urinary incontinence the n. pudendus were cut in female rats, and then MSCs previously differentiated in the myogenic direction were injected into the striated muscle tissue of the urethra. MSCs bound to cells of myogenic lines in vitro, and 4 weeks after injection they contacted with the cells of muscle tissues in vivo. MSCs promoted vascularization and significant recovery of continence, and the sphincter volume was almost restored. Moreover, MSCs were found in the damaged nerve, suggesting a role in nerve repair [20].

A single injection of MSCs partially restores urethral function in an incontinence model. Single as well as repeated doses of 2 × 10^6^ MSCs 1 h, 7 days, and 14 days after vaginal distension and crushing n. pudendus in rats improved the integrity of the urethra, restoring the composition of its connective tissue and neuromuscular structures. MSC treatment improved elastogenesis, prevented dysfunction of the external urethral sphincter, and restored the n. pudendus morphology [7].

Muscle MSCs expressing green fluorescent protein (GFP) were injected into the tail vein of rats 3 days after vaginal overstretching. In samples of the damaged urethra, MSCs were detected only 2 h after injection, but they did not integrate into the tissues. Along with this, MSCs enhance the expression of genes associated with cell proliferation, neural growth factor and extracellular matrix, as well as the expression of smooth and striated muscle proteins in the injured urethra [21].

The muscle and adipose tissue MSC effect on the stress urinary incontinence treatment was investigated experimentally. These cells were isolated from rats and labeled by transfection of an enhanced GFP-gene. Rats received an injection of cells into the bladder neck and transurethral into the sphincter region. Through 0, 15, 30, and 60 days after cell injection the urodynamic test showed that both types of MSCs improved urinary function in rats with internal sphincter deficiency, but the effect of MSC from muscle tissue was more pronounced. According to data of histological analysis in rats treated with MSCs, the content of myosin and α-actin of smooth muscle cells was significantly higher than in the control group after the Hanks solution injection [22].

It was not ruled out that the transplantation of mesenchymal MSCs leads to neovascularization and restoration of muscle cells in animal models of incontinence via the paracrine process [12] or through the exosome production [23,24].

It has been investigated whether local administration of exosomes derived from human adipose tissue MSCs contributes to the correction of experimental stress urinary incontinence in rats. Incontinence was modeled by cutting n. pudendus and vaginal distension. MSCs or exosomes on a Cell Counting Kit-8 matrix were inserted into the peripheral urethra. MSC exosomes can dose-dependently enhance the growth of cultured skeletal muscle and Schwann cell lines. Proteomic analysis has shown that these exosomes contain varied proteins of different signaling pathways: PI3K-Akt, Jak-STAT, and Wnt, which are associated with the regeneration and proliferation of striated muscles and nerves. After exosome injection, rats had a higher bladder capacity and urinary onset pressure, more striated muscle symplasts and peripheral nerve fibers in the urethra than untreated animals. Both urethral function and histological results in rats in the exosome-injected group were slightly better than in the MSC-injected group [23].

Urine-derived MSCs may promote myogenesis after injury of the urethral sphincter muscle (hyperextension of the rat vagina). Following the injection of exosomes from these MSCs the urodynamic parameters of the damaged sphincter have also been improved significantly, and the damaged muscle tissue was restored. The activation, proliferation, and differentiation of own MSC have been proven [24].

Laboratory studies and small sized clinical series suggest applicability of MSCs for the correction of the stress urinary incontinence. However, when summarizing these results, there are often doubts about the adequacy of the impact in modeling experimental pathology. In clinical conditions, the stress urinary incontinence in women is very often diagnosed after vaginal childbirth, which is accompanied by hormonal influences with corresponding changes in the birth canal and surrounding tissues; an initial (congenital) change in the periurethral tissues, including their ligamentous apparatus, is also possible, whereas in experimental modeling, healthy tissues are injured without taking into account their initial state and the general hormonal changes in the body. In addition, it is necessary to note the heterogeneity of recommendations for choosing the best source of MSCs. 

## 5. MSC Using Ineffectiveness and Its Possible Causes 

The data that the cell therapy effectiveness in patients with stress urinary incontinence is lower than expected have emerged recently [10,18,25,26,27]. The main limitations of the MSC using are associated with loss of function after expansion ex vivo, poor engraftment in vivo or survival after transplantation, deposition in the injection site, as well as a lack of understanding of the exact mechanisms of action underlying therapeutic results and MSC behavior in vivo [13].

The results of joint transplantation of muscle-derived cells and mesenchymal MSCs into the urethra have been evaluated. The experiment was carried out on old goats that gave birth many times. The average number of cells with a luminescent label injected per animal was 29.6 ± 4.3 × 10^6^. The urethra samples were obtained at 28 or 84 days after cell transplantation. The transplanted cells were identified in all urethras on day 28, regardless of the type. The most pronounced fluorescence was noted in the co-transplant group. A distinct decrease in the luminescence intensity was observed between 28 and 84 days after all types of transplantation. Both MSCs and muscle cells have promoted striated muscle formation when co-transplanted directly into the external urethral sphincter. These events were rare in the MSC-only group. If cells were injected into the submucosa, they remained undifferentiated, usually in the form of clearly distinguishable repositories. The results showed that co-transplantation of MSCs and muscle cells is more likely to improve urethral closure than transplantation of each cell type separately [10].

The distribution of autologous cells was established during transurethral (under endoscopic control) and periurethral administration to female goats. There have been episodes of leakage of cell suspension in 19% of transurethral injections after needle withdrawal. A repository in the urethral wall was found in all animals 28 days after transplantation. The average percentage of these depots in relation to all injections performed was 68.7% and 67.0% for the groups after periurethral and transurethral injection, respectively. The repository frequency identified in the external urethral sphincter was 18.8% and 17.1%, respectively. Leakage of cell suspension, insufficient accuracy of injection of cells into the external urethral sphincter, and long-term deposition of these cells may contribute to insufficient effectiveness of cell therapy in patients with incontinence [26].

MSC therapy for stress incontinence, common in women with type 2 diabetes and obesity, has also shown low efficacy. It was found that epigenetic changes caused by prolonged exposure to a dyslipidemic environment led to abnormal global transcriptional traits of genes and microRNAs and disrupt the reparative capacity of MSCs in muscle tissue [27].

Literature about efficacy of using MSCs for the correction of stress urinary incontinence reports different results. We speculate that technical variations are largely responsible for the failures and low efficiency of cell therapy, first of all, the speed and method of introducing the cell suspension.

## 6. Conclusions

The cell therapy results in patients and experimental animals with incontinence have shown promising results, but the procedures require further optimization, and more research is needed to focus on the clinical phase. The MSC use appears to be a feasible, safe, and effective method of treatment for patients with urinary incontinence. However, the best mode for application of cell technology is still under study. Most clinical investigations have been performed on only a few patients and during rather short follow-up periods, which, together with an incomplete knowledge of the mechanisms of MSC action, does not make it possible for their widespread implementation. The technical details regarding the MSC application remain to be identified in more rigorous preclinical and clinical trials.

## Data Availability

The data presented in this study are available on request from the corresponding author. The Institute of Chemical Biology and Fundamental Medicine of Siberian Branch of The Russian Academy of Science confirms that article by Maiborodin I. et al. «CELL TECHNOLOGIES IN THE STRESS URINARY INCONTINENCE CORRECTION» fully complies with the legislation of the Russian Federation. The results of literary search can be published in Russian and foreign journals.

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
