# Peer review of "Cell Technologies in the Stress Urinary Incontinence Correction"

_biomedicines, 2022, doi:10.3390/biomedicines10020309_

Round 1

Reviewer 1 Report

The manuscript: ‘Cell technologies in the stress urinary incontinence correction’ is a narrative review about experiments concerning this topic. It mentions clinical and laboratory experiments. The goal of the review is not described, but more importantly, the conclusion introduces some of the problems, that are not mentioned in the core text. I would agree that critique now given in the conclusions is given as an opinion of the authors within the discussion of every experiment-type (every paragraph). That would provide much more clarity. And would justify the conclusion. To state that results are ‘promising’ after discussing the results is not helpful (naïve and meaningless). Furthermore I would like a better sentence for: The urethral sphincter complex and/ or supporting mechanism….. most patients’. A sentence of this is part of the problem, also: how representative are animal experiments that use pudendal lesion, as a model for SUI (long after delivery, where muscles and ligaments were also overstretched/damaged, but indeed including small nerve fibers and blood-vessels). One question, would the knowledge gained from research done for anal sphincter ‘regeneration’ be of help for (stress –urinary) urethral incontinence management. Although it is not unreasonable to summarize preclinical and phase 1 research at a given point, the review as it is now, is not leading to any new insight or helpful to prevent new researches entering dead end streets. Of note, the number of authors of this manuscript is almost larger than the number of experiments that have an applicable conclusion.

Author Response

I am grateful to the Dear Reviewer for the work and time spent in studying our manuscript. Regarding the comments and recommendations made:

The goal of the review is not described

Corrected. Research aim added.

the conclusion introduces some of the problems, that are not mentioned in the core text.

I apologize. Indeed, in "Conclusion" and "Abstract" we have discussed the results too broadly, along with data that is not in the manuscript. At first, the article was called "Cell technologies in urology" and was prepared for the journal "Urology". However, due to the limited volume of manuscripts, the limited number of cited articles and the limited number of co-authors, only part of the manuscript was sent to "Urology". After publication in the journal "Urology" (Maiborodin IV et al. The cell technologies in modification of mesh materials used in urology. Urologiia. 2021;(2):94-99. PMID: 33960166) the excluded material was presented as a separate article for the journal "Biomedicine". None of the paragraphs of both articles are duplicated or the same.

Ttried to correct.

I would agree that critique now given in the conclusions is given as an opinion of the authors within the discussion of every experiment-type (every paragraph).

Tried to correct. Authors' opinions are added at the end of each section.

Furthermore I would like a better sentence for: The urethral sphincter complex and/ or supporting mechanism….. most patients’. A sentence of this is part of the problem, also: how representative are animal experiments that use pudendal lesion, as a model for SUI (long after delivery, where muscles and ligaments were also overstretched/damaged, but indeed including small nerve fibers and blood-vessels).

I agree, this sentence is really unfortunate. My apologies. This sentence has been reformulated.

I also have real doubts about the adequacy of the impact in modeling experimental pathology. In clinical conditions, the stress urinary incontinence in women is very often diagnosed after vaginal childbirth, which is accompanied by hormonal influences with corresponding changes in the birth canal and surrounding tissues; an initial (congenital) change in the periurethral tissues, including their ligamentous apparatus, is also possible. Whereas in experimental modeling, healthy tissues are injured without taking into account their initial state and the general hormonal changes in the body. But this article is in the format of "Literature Review", so the opinion of the authors of the cited articles on the adequacy of the used experimental models is left unchanged.

One question, would the knowledge gained from research done for anal sphincter ‘regeneration’ be of help for (stress –urinary) urethral incontinence management. Although it is not unreasonable to summarize preclinical and phase 1 research at a given point, the review as it is now, is not leading to any new insight or helpful to prevent new researches entering dead end streets.

Apparently, we have not clearly expressed our thoughts somewhere. But we have not written about anal sphincter anywhere.

I beg to differ with the Dear Reviewer that "the review as it is now, is not leading to any new insight or helpful to prevent new researches entering dead end streets". The review will facilitate the selection of the stress urinary incontinence models for future studies. Here shown the main probable reasons for the ineffectiveness of cell therapy, methods of using MSCs in the clinic, etc.

Of note, the number of authors of this manuscript is almost larger than the number of experiments that have an applicable conclusion.

This review is made and written before the start of a new large research topic in which many researchers will participate. I have already noted above that for the first publication in the journal "Urology" only part of the manuscript was sent, and the excluded material was formatted as a separate article for the journal "Biomedicine". In this article, we tried to celebrate all the co-authors who took part in the work.

I hope that we have answered all the questions and comments of the Dear Reviewer, and also clarified the controversial points of our manuscript.

Reviewer 2 Report

Maiborodin et al. provides a review on the use of cell technologies, with a focus of MSCs, in urinary ailments. The review covers everything from in vitro to in vivo findings and provides an up-to-date summary of whats going on.

I have very little input to provide on the science presented in the review. The authors provide excellent background for sections 2-5 but I believe that the authors could easily expand on the experimental details they provide. 

I do however have some major comments regarding the structure and the grammar of the paper and recommend the authors to provide a heavily edited manuscript making the flow of the paper easier to understand as well as editing the grammar of the paper. This is especially needed in the introduction of the review which I had to re-read several times to understand.

Author Response

I want to thank the Dear Reviewer for the short and clear review. On the substance of the remarks made:

I have very little input to provide on the science presented in the review. The authors provide excellent background for sections 2-5 but I believe that the authors could easily expand on the experimental details they provide. 

A large review article was prepared, which was called "Cell technologies in urology" and was send for the journal "Urology". However, due to the limited volume of manuscripts, the limited number of cited articles and the limited number of co-authors, only part of the manuscript was publicized by "Urology". After publication in the journal "Urology" (Maiborodin IV et al. The cell technologies in modification of mesh materials used in urology. Urologiia. 2021;(2):94-99. PMID: 33960166) the excluded material was presented as a separate article for the journal "Biomedicine". None of the paragraphs of both articles are duplicated or the same.

Each paragraph this manuscript is supplemented by the opinion of the authors about the contained data.

I do however have some major comments regarding the structure and the grammar of the paper and recommend the authors to provide a heavily edited manuscript making the flow of the paper easier to understand as well as editing the grammar of the paper. This is especially needed in the introduction of the review which I had to re-read several times to understand.

I can only apologize for the quality of the English language. But the translation was made by a certified translation agency, and then revised by a US citizen.

Once again, I thank the Dear Reviewer for studying the manuscript, and I hope that we have answered all of his questions and comments.

Round 2

Reviewer 1 Report

Consider changing: ‘Aim of the investigation was determined: Based on the analysis of literature data, to study the clinical and experimental results of the use of cellular technologies for the stress urinary incontinence correction.’

…into:  The aim of the investigation was to provide a narrative overview of the literature regarding preclinical and clinical experiments with cell technologies, especially MSCs, to treat stress urinary incontinence.

My Question: Is the aim also: ‘…to improve (or ‘to direct’ or ’to guide’) further research’?

Consider changing: ‘Thus, MSCs increase the content of collagen in the periurethral tissues and persist for a long time in conditions of static stress on several materials used for the manufacture of slings.’ All this is evidence of the promise of using cellular technologies for the stress urinary incontinence correction.

…into: Preclinical studies have demonstrated that MSCs increase the content of collagen* in the periurethral tissues and persist up to 12 weeks in conditions of static stress on several materials used for the manufacture of slings. This may be of relevance for future treatment modalities for SUI.

*But my Question:  is this a sign of dedifferentiation? and What is the opinion of the authors: would it be preferable(better?) that stretch cycling culture is applied, better than continuous stretch, obtain better quality muscle cells?

Consider replacing: ‘The literature contains the results of clinical studies that confirm the effectiveness of even a simple injection of MSCs into the region of urethral sphincters for the stress urinary incontinence correction.’

…with: The results of a small uncontrolled single center clinical study showed effectiveness of injection of MSCs into the region of urethral sphincters for the stress urinary incontinence correction. These results should be confirmed in larger cohort and controlled studies with longer follow up, that also evaluate applicability and safety.

Consider referring to: PMID: 31834471 DOI: 10.1007/s00345-019-03018-9. And or: Nat Rev Urol 2020;17:151–161 (and or: https://www.auajournals.org/doi/10.1097/JU.0000000000001675)

Consider replacing: ‘Numerous experimental results, as well as clinical data, also indicate the high efficiency of cell therapy for the correction of the stress urinary incontinence, that modeled in animals. However, when summarizing these results, there are often doubts about the adequacy of the impact in modeling experimental pathology. In clinical conditions, the stress urinary incontinence in women is very often diagnosed after vaginal childbirth, which is accompanied by hormonal influences with corresponding changes in the birth canal and surrounding tissues; an initial (congenital) change in the periurethral tissues, including their ligamentous apparatus, is also possible. Whereas in experimental modeling, healthy tissues are injured without taking into account their initial state and the general hormonal changes in the body. In addition, it is necessary to note the heterogeneity of recommen[1]dations for choosing the best source of MSCs.’

…with: Laboratory studies and small sized clinical series suggest applicability of MSCs for the correction of the stress urinary incontinence. However, when summarizing these results, there are often doubts about the adequacy of the impact in modeling experimental pathology. In clinical conditions, the stress urinary incontinence in women is very often diagnosed after vaginal childbirth, which is accompanied by hormonal influences with corresponding changes in the birth canal and surrounding tissues; an initial (congenital) change in the periurethral tissues, including their ligamentous apparatus, is also possible. Whereas in experimental modeling, healthy tissues are injured without taking into account their initial state and the general hormonal changes in the body. In addition, it is necessary to note the heterogeneity of recommendations for choosing the best source of MS. Based on our review of the literature we consider …. best applicable.

Consider changing:  ‘Most likely, the inefficiency of using MSCs for the correction of the stress urinary incontinence, noted by some researchers, is due to technical errors in cell therapy.* It is possible** that the increase in the effectiveness of the apply of MSCs is associated with changes in the rate and method of administration, the use of certain scaffolds, as well as changes in cultivation conditions for cells.*

…into: Literature about efficacy of using MSCs for the correction of stress urinary incontinence reports variable results. We speculate that technical variations are largely responsible for this. The rate and method of administration, the use of certain scaffolds, as well as changes in cultivation conditions for cells are all relevant.

However (!):

* My Question: I do not think that you have shown this in the manuscript (or the paragraph), either you show this, or you cannot conclude this.

** is it, or is it not?

Author Response

I want to thank the Dear Reviewer for the very clear review. On the substance of the remarks made:

Consider changing: ‘Aim of the investigation was determined: Based on the analysis of literature data, to study the clinical and experimental results of the use of cellular technologies for the stress urinary incontinence correction.’

…into:  The aim of the investigation was to provide a narrative overview of the literature regarding preclinical and clinical experiments with cell technologies, especially MSCs, to treat stress urinary incontinence.

My Question: Is the aim also: ‘…to improve (or ‘to direct’ or ’to guide’) further research’?

I totally agree. The aim of the investigation has been changed almost in accordance with the recommendations of the Dear Reviewer.

Consider changing: ‘Thus, MSCs increase the content of collagen in the periurethral tissues and persist for a long time in conditions of static stress on several materials used for the manufacture of slings.’ All this is evidence of the promise of using cellular technologies for the stress urinary incontinence correction.

…into: Preclinical studies have demonstrated that MSCs increase the content of collagen* in the periurethral tissues and persist up to 12 weeks in conditions of static stress on several materials used for the manufacture of slings. This may be of relevance for future treatment modalities for SUI.

*But my Question:  is this a sign of dedifferentiation? and What is the opinion of the authors: would it be preferable(better?) that stretch cycling culture is applied, better than continuous stretch, obtain better quality muscle cells?

I agree. The sentence has been modified almost in accordance with the recommendations.

Increase the content of collagen cannot be a sign of MSC dedifferentiation. Collagen is produced by fibroblasts, and MSCs either differentiate into fibroblasts (Maiborodin I.V. et al. Possibility of aggravation of tissue sclerosis after injection of multipotent mesenchymal stromal cells near the forming cicatrix in the experiment. // Bull. Exp. Biol. Med. – 2017. - Vol. 163. - â„– 4. – P. 554–560. DOI 10.1007/s10517-017-3848-1.) or initiate an increase in collagen synthesis by "old" fibroblasts due to direct influence or through exposure by paracrine methods, via exosomes.

I am not qualified to judge which culture method is best for increasing the quality of muscle cells. But I think stretch cycling is better. When cultivating MSCs under conditions of prolonged static tension of the fabric matrix, the cells simply cover the entire surface of the textile in an level layer and, possibly, unstick from it upon sudden stretching, even without trypsin. As a result of MSC growth under conditions of cyclic contraction and stretching of the textile scaffold, the cells will constantly stretch and contract together with the fabric; it can be said that growing MSCs will be "trained" in the process of differentiation into muscle cells.

Consider replacing: ‘The literature contains the results of clinical studies that confirm the effectiveness of even a simple injection of MSCs into the region of urethral sphincters for the stress urinary incontinence correction.’

…with: The results of a small uncontrolled single center clinical study showed effectiveness of injection of MSCs into the region of urethral sphincters for the stress urinary incontinence correction. These results should be confirmed in larger cohort and controlled studies with longer follow up, that also evaluate applicability and safety.

Corrected

Consider referring to: PMID: 31834471 DOI: 10.1007/s00345-019-03018-9. And or: Nat Rev Urol 2020;17:151–161 (and or: https://www.auajournals.org/doi/10.1097/JU.0000000000001675)

Thank you. I know these works, the article by C.J. Hillary [new 18] was already present among the References. But I try to use the results of original research as much as possible when creating a review of the literature. I have included a link to M.R. Kaufman into the manuscript.

Consider replacing: ‘Numerous experimental results, as well as clinical data, also indicate the high efficiency of cell therapy for the correction of the stress urinary incontinence, that modeled in animals. However, when summarizing these results, there are often doubts about the adequacy of the impact in modeling experimental pathology. In clinical conditions, the stress urinary incontinence in women is very often diagnosed after vaginal childbirth, which is accompanied by hormonal influences with corresponding changes in the birth canal and surrounding tissues; an initial (congenital) change in the periurethral tissues, including their ligamentous apparatus, is also possible. Whereas in experimental modeling, healthy tissues are injured without taking into account their initial state and the general hormonal changes in the body. In addition, it is necessary to note the heterogeneity of recommen[1]dations for choosing the best source of MSCs.’

…with: Laboratory studies and small sized clinical series suggest applicability of MSCs for the correction of the stress urinary incontinence. However, when summarizing these results, there are often doubts about the adequacy of the impact in modeling experimental pathology. In clinical conditions, the stress urinary incontinence in women is very often diagnosed after vaginal childbirth, which is accompanied by hormonal influences with corresponding changes in the birth canal and surrounding tissues; an initial (congenital) change in the periurethral tissues, including their ligamentous apparatus, is also possible. Whereas in experimental modeling, healthy tissues are injured without taking into account their initial state and the general hormonal changes in the body. In addition, it is necessary to note the heterogeneity of recommendations for choosing the best source of MS. Based on our review of the literature we consider …. best applicable.

Corrected, except for the last sentence. Most researchers in the urology clinic offer MSCs derived from adipose tissue. These cells (autocells!) are the easiest to isolate from a patient. However, based on my own experience, I believe that MSCs of bone marrow origin are the most effective.

Consider changing:  ‘Most likely, the inefficiency of using MSCs for the correction of the stress urinary incontinence, noted by some researchers, is due to technical errors in cell therapy.* It is possible** that the increase in the effectiveness of the apply of MSCs is associated with changes in the rate and method of administration, the use of certain scaffolds, as well as changes in cultivation conditions for cells.*

…into: Literature about efficacy of using MSCs for the correction of stress urinary incontinence reports variable results. We speculate that technical variations are largely responsible for this. The rate and method of administration, the use of certain scaffolds, as well as changes in cultivation conditions for cells are all relevant.

However (!):

* My Question: I do not think that you have shown this in the manuscript (or the paragraph), either you show this, or you cannot conclude this.

** is it, or is it not?

I agree. The sentence has been modified almost in accordance with the recommendations. Indeed, the manuscript actually contains a description of only technical problems as the reasons for the ineffectiveness of cell therapy. The manuscript has been modified accordingly.

Once again, I thank the Dear Reviewer for thoughtful studying the manuscript, and I hope that we have answered all of his questions and comments.

Reviewer 2 Report

The authors have revised the manuscript considerably and I have no further comments 

Author Response

The authors have revised the manuscript considerably and I have no further comments 

I am grateful to the Dear Reviewer for the work and time spent in studying our manuscript.

Round 3

Reviewer 1 Report

none